# GEMCONT: Genetics-based Multimodal Contrastive Learning Enhances Phenotypic embeddings and Boosts Genetic Discovery

**Daniel Sens**[1,2,3]            DANIEL.SENS@HELMHOLTZ-MUNICH.DE
**Liubov Shilova**[1,2,3,4]        LIUBOV.SHILOVA@HELMHOLTZ-MUNICH.DE
**Adrian V. Dalca**[5,6]             ADALCA@MIT.EDU
**Julia A. Schnabel**[3,7,8,*]      JULIA.SCHNABEL@HELMHOLTZ-MUNICH.DE
**Francesco Paolo Casale**[1,2,3,*]   FRANCESCOPAOLO.CASALE@HELMHOLTZ-MUNICH.DE

[1] *Institute of AI for Health, Helmholtz Zentrum München – German Research Center for Environmental Health, Neuherberg, Germany*

[2] *Helmholtz Pioneer Campus, Helmholtz Zentrum München – German Research Center for Environmental Health, Neuherberg, Germany*

[3] *School of Computation, Information and Technology, Technical University of Munich, Garching, Germany*

[4] *Friedrich-Alexander-Universität Erlangen-Nürnberg, Erlangen, Germany*

[5] *A.A. Martinos Center for Biomedical Imaging, Massachusetts General Hospital and Harvard Medical School, Boston, MA, USA*

[6] *Computer Science and Artificial Intelligence Laboratory, Massachusetts Institute of Technology, Cambridge, MA, USA*

[7] *Institute of Machine Learning in Biomedical Imaging, Helmholtz Zentrum München – German Research Center for Environmental Health, Neuherberg, Germany*

[8] *School of Biomedical Engineering and Imaging Sciences, King's College London, London, UK*

**Editors:** Accepted for publication at MIDL 2026

## Abstract

Genetic variation provides stable, time-invariant markers of disease risk and can therefore reveal upstream mechanisms underlying complex traits. Genome-wide association studies (GWAS) have identified thousands of loci associated with disease, yet most remain difficult to interpret because the intermediate phenotypes linking genotype to disease are unknown. Here, we address the question whether disease-associated genetic loci can be directly used to extract such risk-related features from quantitative phenotypes, including functional tests and medical imaging. We introduce **GEMCONT** (GEnetics-based Multimodal CONTrastive Learning), a multimodal contrastive learning framework that aligns genotype and phenotype representations in a shared latent space. Unlike task-agnostic multimodal pretraining, GEMCONT is disease-conditioned: GWAS-informed variant panels act as targeted supervision to learn risk-relevant imaging embeddings. To reflect the weak, additive nature of genetic effects, it employs a linear genetic encoder alongside a deep phenotypic encoder. We validate GEMCONT in controlled simulations and apply it to two real-world settings: spirometry curves for asthma and retinal fundus images for glaucoma. In both, GEMCONT improves disease risk prediction and enhances recovery of genetic associations compared with standard unsupervised or polygenic risk–based models.

---

[*] Corresponding authors.

Altogether, our results demonstrate that incorporating stable genetic supervision into multimodal representation learning enables the extraction of genetically informed risk traits, refining disease phenotypes and improving the interpretability of association studies.

**Keywords:** Multimodal Contrastive Learning, Imaging Genetics, Genome-Wide Association Studies, Machine Learning–Derived Phenotypes, Medical Imaging

## 1. Introduction

Genome-wide association studies (GWAS) have identified thousands of genetic loci associated with human diseases and complex traits (Visscher et al., 2017; Manolio et al., 2009). Because germline variation is fixed and precedes disease onset, genetic associations provide upstream information about biological mechanisms. Yet for most loci, the functional link between genotype and phenotype remains unknown (Tam et al., 2019). Recent work in imaging genetics has begun to address this challenge by using high-dimensional biomedical data—such as medical images or physiological recordings—to derive quantitative phenotypes that better reflect underlying biology (Wright and Herzberg, 2021; Tracy, 2008; Robinson, 2012). Early approaches relied on manually defined regions or handcrafted measurements, while more recent studies leverage machine learning to learn compact phenotypic representations directly from raw data (Zech et al., 2018). These representations have proven valuable for association studies: for instance, supervised networks trained on clinical outcomes can uncover novel genetic loci (Rakowski et al., 2024; Kirchler et al., 2022), and unsupervised embeddings can reveal heritable structure (Yun et al., 2024; Xie et al., 2024). However, unsupervised representation learning tends to capture the dominant axes of variation in the data and may overlook disease-related effects when these correspond to more subtle or less frequent patterns (Shilova et al., 2025).

To address these challenges, we introduce **GEMCONT** (GEnetics-based Multimodal CONTrastive learning), a multimodal contrastive learning framework for imaging-genetics analysis (Fig. 1). GEMCONT aligns medical imaging data with disease-associated genetic variants in a shared latent space for a given disease. Through this alignment, GEMCONT learns disease-specific imaging embeddings that capture risk-relevant variation and are predictive of future disease. By focusing on disease-specific, risk-aligned embeddings, GEMCONT differs from prior multimodal frameworks such as ContIG (Taleb et al., 2022) and MRM (Yang et al., 2023), which use genetic or molecular modalities for task-agnostic multimodal pretraining followed by downstream fine-tuning. The contributions of this work are threefold:

1. **Genetics-informed contrastive learning.** We adapt multimodal contrastive learning to disease-focused imaging-genetics analysis through GEMCONT, which (i) selects disease-associated variants from external GWAS summary statistics to define targeted genetic supervision, and (ii) employs a linear genetic projector for efficient and interpretable variant contributions (Fig. 1).

2. **Benchmarking across genetic architectures.** We benchmark GEMCONT using controlled simulations with known disease-associated latent traits and causal variants, evaluating performance across sample sizes and genetic architectures to determine when contrastive alignment improves latent trait recovery.

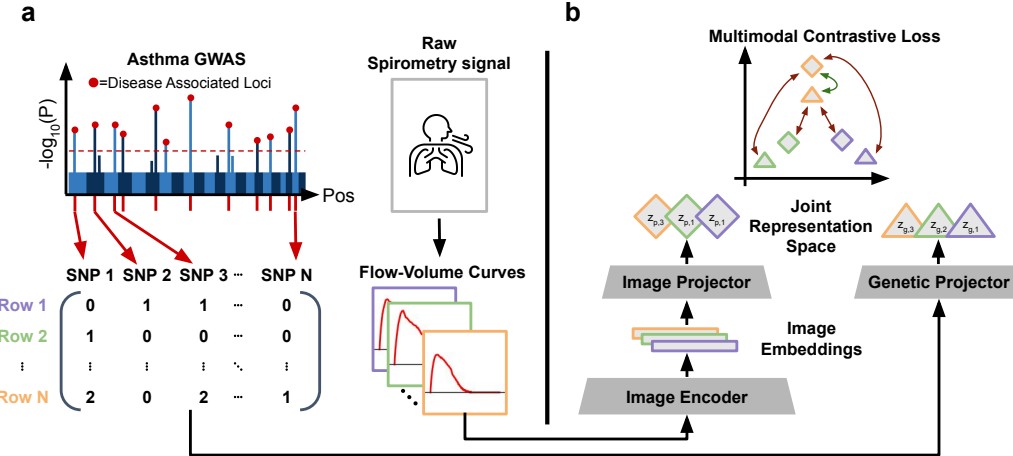

Figure 1: **Application of GEMCONT to asthma variants and spirometry images.**
**(a)** Genetic variants significantly associated with asthma are extracted from
the imputed UK Biobank data. **(b)** Corresponding raw spirometry signals are
converted into Flow-Volume Curve images, and an asymmetric dual-encoder is
trained to align genotype and image embeddings in a joint representation space
through the multimodal contrastive loss.

3. **Applications to population imaging.** We apply GEMCONT to two use cases in
the UK Biobank (Sudlow et al., 2015). First, we recover asthma-related spirometry
embeddings by integrating asthma-associated variants with flow–volume curve data.
Second, we recover a glaucoma-related latent trait from retinal fundus images using
glaucoma-associated variants and imaging data. In both settings, genetics-guided
contrastive learning improves disease risk prediction and strengthens genetic association
analyses compared to standard self-supervised approaches.

Together, these results establish contrastive learning with genetic supervision as a principled
approach for constructing disease-specific, GWAS-aware phenotypes from high-dimensional
medical imaging data.

## 2. Methods

The approach implemented in GEMCONT can be formulated as two-step process: (i) learning
a joint representation space where genetic and imaging-derived features are co-embedded,
and (ii) validating the extracted imaging embeddings in statistical association analyses and
disease classification. Below, we describe the co-embedding pipeline and the statistical
validation procedures.

### 2.1. Contrastive Learning for Genetics-Image Alignment

Contrastive learning for genetics-imaging alignment builds on the CLIP (Radford et al., 2021) framework, introducing key adaptations to address the unique challenges of genetic data, which are characterized by sparse and weak additive effects on phenotypes. Formally, given a dataset of paired genotype and phenotype samples $\mathcal{D} = \{(x_{g,i}, x_{p,i})\}_{i=1}^{N}$, each genotype sample $x_g \in \{0, 1, 2\}^S$ represents an allele count vector of $S$ disease-associated variants, while $x_p$ denotes a high-content phenotype (e.g., medical images). Following the principle implemented in the CLIP model, GEMCONT learns joint embeddings of genetic and imaging data by maximizing agreement between modalities from the same individual while encouraging separation between individuals. This objective enables the co-embedding of genetic and phenotypic features into a shared latent space, facilitating the discovery of biologically meaningful genotype-phenotype relationships.

**Multimodal Contrastive Learning Objective.** GEMCONT processes genotype and phenotype data through modality-specific encoders (Fig. 1). The phenotype encoder $f_{\theta_p}$ maps $x_p$ to an intermediate embedding $e_p$, which is then projected onto a unit-norm latent space as $z_p$. The genotype data are processed through a linear genotype projector, mapping $x_g$ to a unit-norm latent embedding $z_g$. During training, we sample a mini-batch $\mathcal{B} \subset \mathcal{D}$ of paired genotype–phenotype samples and optimize the phenotype encoder, phenotype projector, and genotype projector to align genetic and phenotypic projections by maximizing similarity within individuals while minimizing it across individuals. This is achieved using the multimodal contrastive loss (Radford et al., 2021):

$$\mathcal{L} = \frac{1}{2}\left(\mathcal{L}_{g\to p} + \mathcal{L}_{g\leftarrow p}\right), \tag{1}$$

where

$$\mathcal{L}_{g\to p} = -\sum_{j\in\mathcal{B}} \log \frac{\exp(z_{g,j}^T z_{p,j}/\tau)}{\sum_{k\in\mathcal{B},\, k\neq j} \exp(z_{g,j}^T z_{p,k}/\tau)}, \tag{2}$$

and $\mathcal{L}_{g\leftarrow p}$ is defined analogously, swapping $g$ and $p$. Similar to CLIP, $\tau > 0$ is a learnable temperature parameter.

**Adaptations for genetic data.** To address the sparsity and additive nature of genetic effects, GEMCONT introduces two key adaptations:

1. **Selection of informative variants.** We extract relevant genetic features from genome-wide association study (GWAS) summary statistics, which quantify associations (e.g., p-values, effect sizes) between millions of variants and a disease of interest. From these statistics, we select independent genome-wide significant loci through a clumping procedure (Purcell et al., 2007), iteratively retaining the most significant variant while removing correlated neighbors within a 5 Megabase (Mb) window.

2. **Linear projection of genotypes.** Genetic effects are predominantly additive with limited evidence for interactions between variants (Hill et al., 2005), making a linear projection sufficient for mapping selected variants into the latent space. This reduces model complexity while preserving key genetic signals.

## 2.2. Genetic Association Analysis of Learned Embeddings

**Multi-trait GWAS for embedding analysis**. To assess whether GEMCONT-derived embeddings capture meaningful genetic signals, we perform a multi-trait genome-wide association study (GWAS) on a held-out set of samples not used for training. We adapt the single-variant model from (Lippert et al., 2014; Casale et al., 2015), modeling each embedding dimension as a quantitative trait influenced by genetic variation. Let $\boldsymbol{E} \in \mathbb{R}^{N \times D}$ be the embedding matrix, $\boldsymbol{g} \in \{0, 1, 2\}^{N \times 1}$ a genotype vector, and $\boldsymbol{F} \in \mathbb{R}^{N \times K}$ a matrix of covariates. The model is:

$$\boldsymbol{E} = \boldsymbol{F}\boldsymbol{A} + \boldsymbol{g}\boldsymbol{b}^T + \boldsymbol{\Psi}, \tag{3}$$

where $\boldsymbol{A} \in \mathbb{R}^{D \times K}$ and $\boldsymbol{b} \in \mathbb{R}^{D \times 1}$ capture covariate and genetic effects, respectively, and $\boldsymbol{\Psi} \sim \mathcal{N}(0, \boldsymbol{C})$ models residual noise with a learnable covariance matrix $\boldsymbol{C} \in \mathbb{R}^{D \times D}$. Following (Lippert et al., 2014; Casale et al., 2015), $\boldsymbol{C}$ is estimated under the null model, while a single scaling factor per variant is optimized under the alternative model to control false positives efficiently (Korte et al., 2012). We test whether $\boldsymbol{b} \neq 0$, obtaining p-values via a likelihood ratio test with $D$ degrees of freedom. For simulations (Sec. 3.1), we do not adjust for covariates. In both real-world applications (Sec. 3.2, Sec. 3.3), all embedding-GWAS tests adjust for genotyping array, assessment center, sex, age, age$^2$, sex-by-age, sex-by-age$^2$, height, height$^2$, BMI, and the top 20 genetic principal components. To address feature correlation and non-Gaussian distributions, embeddings are projected onto their top $D$ principal components and rank-normalized before association testing. Independent genome-wide significant loci ($p < 5 \times 10^{-8}$) are identified using PLINK's clumping procedure (Purcell et al., 2007), which retains only approximately independent genetic associations by removing variants in linkage disequilibrium ($r^2 < 0.05$) within a 5 Mb window.

**Assessing overlap with disease GWAS**. To evaluate whether GEMCONT-derived embeddings capture known disease-associated genetic signals, we compare the genomic loci identified in our embedding-based GWAS to those from a standard disease GWAS. Specifically, we measure the fraction of independent genome-wide significant loci ($p < 5 \times 10^{-8}$) identified in the disease GWAS that are also recovered at genome-wide significance in the embedding GWAS. This assessment is performed on a held-out test set, distinct from the training data used for learning embeddings.

**External disease GWAS and meta-analysis.** To define the disease-specific variant panels used by GEMCONT, we rely on large external genome-wide association studies (GWAS) for asthma and glaucoma from the Million Veteran Program (Verma et al., 2023) and FinnGen (Kurki et al., 2023). For each disease, we harmonize summary statistics across cohorts and combine them using a fixed-effect inverse-variance meta-analysis in METAL (Willer et al., 2010). From the resulting meta-analytic GWAS, we selected variants with association $p < 10^{-5}$ and applied LD clumping in PLINK (5 Mb window, $r^2 < 0.05$) (Purcell et al., 2007), yielding an approximately independent set of disease-enriched SNPs. The $10^{-5}$ threshold is an intermediate cut-off that has been used when selecting variants from GWAS loci for Mendelian randomization analyses (Davey Smith and Hemani, 2014; Jin et al., 2024) and lies within the range of $p$-value thresholds typically explored in clumping-and-thresholding polygenic risk score methods (Choi et al., 2020). This choice provides a panel of variants that is strongly enriched for disease-associated signal while remaining sufficiently large to supervise the phenotype encoder.

## 3. Experiments and Results

We evaluate GEMCONT's ability to (i) enhance genetic signal for phenotype- or disease-associated variants and (ii) recover the underlying latent phenotype in simulations, and via disease risk prediction as a proxy in real data applications. We conduct three experiments: a controlled simulation study to assess performance under varying genetic architectures and two real-world applications to UK Biobank (Sudlow et al., 2015) data. In the first application we analyze flow-volume curves - used in asthma diagnosis (Jayasooriya et al., 2023) - and integrate genetic variants associated with asthma. In the second we apply our framework to retina fundus images, which are used in glaucoma diagnosis (Saha et al., 2023), and co-embed them with variants associated with glaucoma. We additionally report robustness analyses comparing GEMCONT with linear versus nonlinear genetic projectors and evaluating sensitivity to the GWAS variant-selection threshold (Sec. 3.4; Tab. 1).

**Compared methods.** In the simulation and spirometry experiments, we compare GEMCONT against two established self-supervised embedding methods: a variational autoencoder (VAE), which learns latent representations by optimizing a reconstruction objective under a latent prior (Kingma and Welling, 2014), and SimCLR (Chen et al., 2020), a contrastive learning approach that maximizes agreement between augmented views of the same input. For the fundus application, we leverage the retina foundation model RetFound (Zhou et al., 2023, 2025), a large-scale self-supervised model pretrained on nearly one million fundus images, which provides a strong unsupervised reference without the need to retrain SimCLR or VAE baselines. In the real-data applications, we additionally include baselines tailored to disease prediction. First, we use a simple multimodal model in which the genetic branch is reduced to a single polygenic risk score (PRS) for the target disease, computed as the sum of GEMCONT's input variants weighted by their GWAS effect sizes and fed as a univariate input to the genetic projector. We refer to this as the PRS baseline. Second, to assess whether genetics-driven phenotype embeddings provide added value over conventional clinical markers, we benchmark all spirometry models against the $FEV_1/FVC$ ratio and all fundus models against the cup-to-disc ratio, both widely used functional (Lambert et al., 2015) and imaging-derived (Gordon et al., 2002; Foster et al., 2002) biomarkers in their respective diagnostic domains. Finally, in the fundus experiment we leverage a strong retinal foundation model (a DINOv2-pretrained ViT backbone) and consider three configurations on top of it: a frozen RetFound (Zhou et al., 2023, 2025) baseline, GEMCONT, and a supervised upper-bound model (Sec. 3.3).

### 3.1. Benchmarking in Simulated Data

**Setup.** To evaluate GEMCONT in a controlled setting, we design a simulation framework where a latent genetic trait influences imaging features, mimicking real-world genotype-phenotype relationships. We use EMNIST (Cohen et al., 2017), a dataset of 814,255 grayscale handwritten characters across 62 classes, and define the latent phenotype as the rotation angle of each character. We systematically vary key factors: training set size ($N_{train}$), genetic variance explained ($h_g$), and phenotype transformation strength ($\alpha_{max}$). A fixed 100K test set is used across all experiments, with five random splits for training, where $N_{train}$ is varied (default: 100K). After training, we extract image embeddings and evaluate:

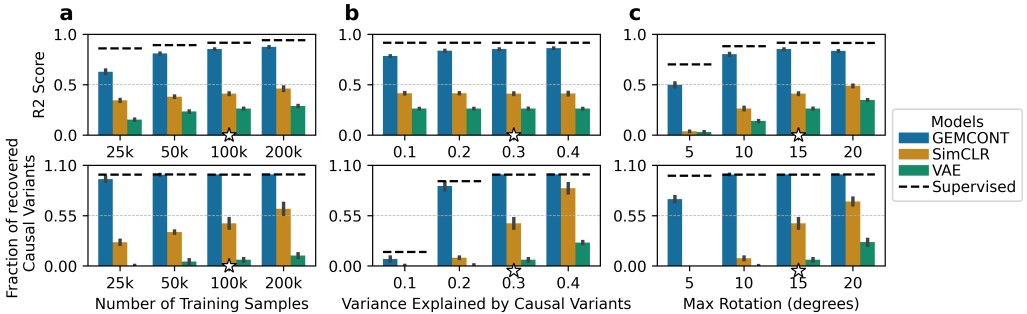

Figure 2: **Evaluation of phenotype and genetic signal recovery.** Performance of GEMCONT, SimCLR, VAE, and a supervised model is assessed under varying training set sizes (a), genetic variance explained (b), and maximum rotation (c). The first row shows the $R^2$ score for predicting the latent phenotype $z$ from the learned embeddings, while the second row presents the fraction of causal variants identified at genome-wide significance ($p < 5 \times 10^{-8}$). Each bar represents the mean $\pm$ standard deviation across five random splits. Stars denote standard values held constant while other parameters were varied.

1. **Phenotype recovery:** Predicting $z$ from embeddings using ridge regression, measured by $R^2$ on the test set.

2. **Genetic recovery:** Identifying genome-wide significant variants ($p < 5 \times 10^{-8}$) via multi-trait GWAS (Sec. 2.2).

**Simulation strategy.** We first subsample $N$ images stratified across character labels. Next, we simulate a genotype matrix $\boldsymbol{G} \in \{0, 1, 2\}^{N \times S}$ for $S$ variants, where each entry represents allele counts drawn from a binomial distribution: $\boldsymbol{G}_{i,j} \sim \text{Binomial}(2, f_j)$ with minor allele frequency $f_j$. The latent phenotype $\boldsymbol{z}$ is then generated as a weighted combination of genetic effects and environmental noise, controlling the proportion of variance explained by genetics ($h_g$):

$$\boldsymbol{z} = \sqrt{h_g} \cdot \widetilde{\boldsymbol{G}\boldsymbol{\beta}} + \sqrt{1 - h_g} \cdot \widetilde{\boldsymbol{z}_n}, \tag{4}$$

where $\widetilde{\boldsymbol{x}} = \frac{\boldsymbol{x} - \mu(\boldsymbol{x})}{\sigma(\boldsymbol{x})}$ denotes the standardized version of a vector $\boldsymbol{x}$, with $\mu(\cdot)$ and $\sigma(\cdot)$ denoting the mean and standard deviation of its elements, respectively. Here, each variant effect size $\boldsymbol{\beta}$ is sampled from $\{-1, 1\}$ with equal probability, and $\boldsymbol{z}_n \sim \mathcal{N}(\boldsymbol{0}, \boldsymbol{I}_N)$ models environmental noise. We then define rotation angles as $\boldsymbol{\alpha} = \alpha_{\max} \cdot \tanh(c \cdot \boldsymbol{z})$, where $c$ is chosen to prevent saturation of tanh across samples. Each image is rotated by its corresponding $\alpha_j$, creating a dataset of genetic-image pairs directly linked through rotation.

**Results.** Figure 2 summarizes the results. GEMCONT outperforms SimCLR and VAE across all settings. Performance saturates beyond 100K training samples, though GEMCONT maintains a significant advantage (Fig. 2a). Genetic variance ($h_g$) has minimal impact on phenotype recovery but strongly affects variant detection, with GEMCONT consistently identifying more causal variants (Fig. 2b). Finally, lower rotation angles ($\alpha_{\max}$) degrade baseline performance more than GEMCONT, which remains robust across conditions

(Fig. 2c). As expected, a supervised model serves as an upper bound for both phenotype and variant recovery.

### 3.2. Application to Spirometry and Asthma

**Experimental setup.** We generate flow–volume curves following (Yun et al., 2024) and compute the $FEV_1/FVC$ ratio, a key biomarker for asthma diagnosis (Lambert et al., 2015). Similar to recent work that applies CNNs directly to images of spirometry flow–volume curves for quality control (Martins et al., 2025; Wang et al., 2022), we rasterize each trajectory into a standardized $256 \times 256$ grayscale image at 200 dpi and use these images as input to the encoder. Genetic variants associated with asthma are selected from external GWAS summary statistics using clumping (Sec. 2.2), yielding 551 approximately independent variants. To ensure that spirometry curves reflect baseline lung function, we exclude participants who reported using a chest inhaler or smoking a cigarette within the last hour before testing, in line with clinical spirometry preparation guidelines (Paraskeva et al., 2011). After matching imaging and genetic data and restricting to individuals of European ancestry, we retain 227,332 participants. To obtain stable estimates and quantify variability, we perform five random 50/50 train/validation splits and evaluate (i) disease recovery and (ii) genetic signal enrichment in the embeddings. Disease recovery is assessed using L2-regularized logistic regression to predict asthma from pre-spirometry diagnosis and diagnosis within five years post-assessment, reporting ROC AUC. Genetic signal enrichment is quantified by performing multi-trait GWAS on the embeddings and computing the fraction of independent asthma-associated variants that remain significant after Bonferroni correction (Sec. 2.2). Figure 3 summarizes the results.

**Results.** GEMCONT achieves the highest recall of asthma-associated loci, though all models recover only a small fraction (Fig. 3a), consistent with expectations given our smaller sample size relative to the effective sample size of the GWAS meta-analysis. For asthma prediction, GEMCONT significantly ($p < 0.05$) outperforms the compared models at baseline and approaches the supervised model's upper bound for future diagnoses (Fig. 3b). Finally, Fig. 3c displays violin plots of the first principal component of the image embeddings (PC1) and the $FEV_1/FVC$ ratio, stratified by asthma status; both measures show modest distributional shifts between cases and controls, indicating that PC1 captures asthma-related variation that is comparable in magnitude to the classical biomarker but derived directly from the flow–volume curves.

### 3.3. Application to Fundus Images and Glaucoma

**Experimental setup.** We analyzed color fundus images from the first imaging visit of UK Biobank participants (Sudlow et al., 2015) and filtered images using the MCF-Net model (Fu et al., 2019), excluding images with a rejection probability above 80%. We further excluded fundus images from participants who reported prior surgery or laser treatment for glaucoma, as this affects biomarkers for glaucoma risk and can impact fundus morphology (Lesk et al., 1999; Raghu et al., 2012; Pillunat et al., 2023). Genetic variants associated with glaucoma were selected from external GWAS summary statistics using clumping (Sec. 2.2), yielding 1,535 approximately independent variants. After merging with genetic data for individuals of European ancestry, we retained 36,349 participants with at least one usable fundus image.

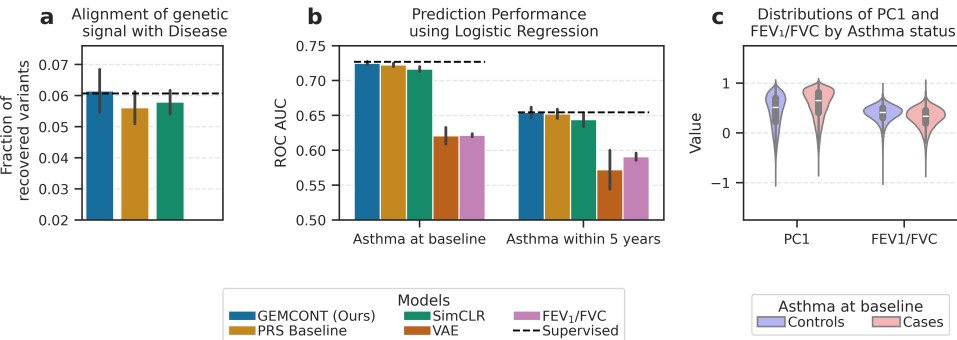

Figure 3: **Asthma prediction and genetic signal enrichment.** Comparison of GEM-CONT, PRS baseline, SimCLR, VAE, and a supervised model trained on the latent phenotype. (a) Fraction of independent genome-wide significant asthma-associated variants recovered in method-specific GWAS after multiple testing correction. (b) ROC AUC for asthma classification at baseline and within 5 years post-assessment. (c) Distributions of the first principal component (PC1) of GEMCONT image embeddings and of $FEV_1/FVC$, stratified by asthma status. Violin plots show normalized values for controls (blue) and cases (red), with black bars indicating median and inter-quartile range. Results are mean $\pm$ standard deviation across 5 random 50/50 splits.

For individuals with two images, we randomly sampled left or right eye with equal probability during training whenever the individual was drawn into a batch. During validation, if both eyes were available, we extracted image embeddings for each eye and used their mean as the final embedding. We build on a Vision Transformer (ViT) base encoder pretrained using DINOv2 on retinal images (Zhou et al., 2023, 2025), which we keep frozen and use as an online feature extractor during training (Kolesnikov et al., 2020; Vo et al., 2025). Standard image augmentations are applied before feeding inputs through the frozen encoder (Sec. 3.5). On top of this backbone, we consider three image-based configurations. First, RetFound uses the frozen ViT features with simple mean pooling over patch tokens; no additional representation learning is performed, and the resulting embeddings are used directly in downstream GWAS and logistic regression. Second, GEMCONT adds an attention-pooling layer (Ilse et al., 2018) and a lightweight two-layer MLP to map pooled features to image embeddings, which are then aligned with genetic embeddings using the multimodal contrastive objective. Third, a supervised model shares the same architecture as GEMCONT but is optimized directly for glaucoma classification, providing an approximate upper bound. As in the spirometry experiment (Sec. 3.2), we perform five random 50/50 train/validation splits and evaluate (i) disease recovery and (ii) genetic signal enrichment in the embeddings. Disease recovery is assessed using L2-regularized logistic regression to predict glaucoma from diagnosis by the time of image acquisition and diagnosis within five years post-assessment, reporting ROC AUC. Genetic signal enrichment is quantified via multi-trait GWAS on the embeddings, measuring the fraction of independent glaucoma-associated variants that remain significant after Bonferroni correction (Sec. 2.2). Figure 4 summarizes the results.

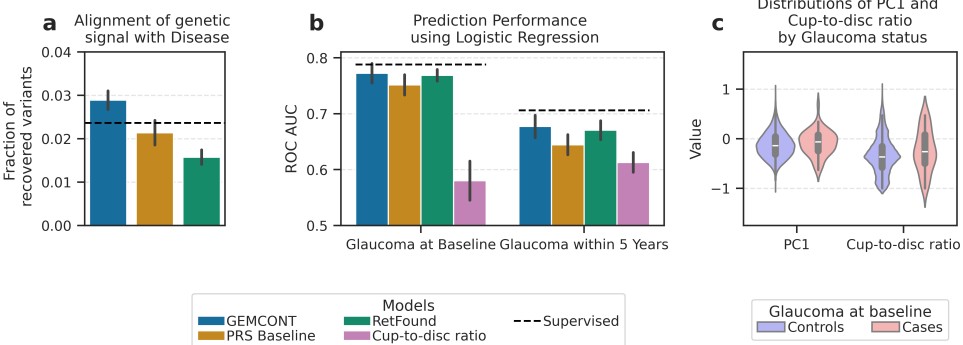

Figure 4: **Glaucoma prediction and genetic signal enrichment.** Comparison of GEM-CONT, a PRS-only baseline, RetFound embeddings, a supervised upper-bound model (dashed line), and the cup-to-disc ratio clinical biomarker. (a) Fraction of independent genome-wide significant glaucoma-associated variants recovered after multiple testing correction. (b) ROC AUC for glaucoma classification at baseline and within 5 years post-assessment. (c) Distributions of the first principal component (PC1) of GEMCONT image embeddings and of cup-to-disc ratio, stratified by glaucoma status. Violin plots show normalized values for controls (blue) and cases (red), with black bars indicating median and inter-quartile range. Results are mean ± standard deviation across 5 random 50/50 splits.

**Results.** GEMCONT recovers the largest fraction of independent glaucoma-associated variants, outperforming the PRS and RetFound baselines as well as the supervised upper bound in terms of alignment between embeddings and disease loci (Fig. 4a). For disease prediction, GEMCONT is competitive with the supervised model and consistently outperforms both the PRS baseline and the cup-to-disc ratio for glaucoma at image acquisition and for incident glaucoma within five years (Fig. 4b). Finally, both the first principal component of the GEMCONT embeddings and the cup-to-disc ratio show case–control shifts, with PC1 exhibiting slightly stronger separation, indicating that the learned representation captures glaucoma-related variation that is at least comparable to this established imaging biomarker (Fig. 4c).

### 3.4. Robustness and Sensitivity Analyses

We conducted robustness analyses to evaluate two key design choices: (i) the use of a linear genetic projector and (ii) the GWAS variant-selection threshold. Replacing the linear genetic projector with a two-layer MLP (batch normalization + ReLU) did not produce consistent gains in disease-prediction AUC or genetic-signal recovery across asthma and glaucoma, supporting the linear inductive bias under weak and sparse genetic effects (Tab. 1). Likewise, applying a stricter GWAS-significant supervision panel ($p < 5 \times 10^{-8}$, LD-clumped) resulted in comparable or slightly lower AUC and no systematic improvement in recovery relative to the default $p < 10^{-5}$ panel (Tab. 1).

Table 1: **GEMCONT ablation and sensitivity analyses on asthma (spirometry) and glaucoma (fundus).** ROC AUC is reported for baseline diagnosis and diagnosis within 5 years. Genetic recovery is the fraction of independent loci from the corresponding external disease GWAS recovered as significant in embedding-GWAS at $p < 5 \times 10^{-8}$ (held-out data). Values are mean $\pm$ std across 5 random splits.

| Experiment | Method (setting) | ROC AUC | | Genetic recovery |
|---|---|---|---|---|
| | | baseline | within 5 years | |
| Asthma (spirometry) | GEMCONT | $0.725 \pm 0.002$ | $0.654 \pm 0.008$ | $0.062 \pm 0.007$ |
| | GEMCONT (MLP) | $0.723 \pm 0.003$ | $0.654 \pm 0.008$ | $0.060 \pm 0.005$ |
| | GEMCONT (strict SNP panel) | $0.723 \pm 0.002$ | $0.652 \pm 0.008$ | $0.057 \pm 0.005$ |
| Glaucoma (fundus) | GEMCONT | $0.772 \pm 0.017$ | $0.677 \pm 0.020$ | $0.029 \pm 0.002$ |
| | GEMCONT (MLP) | $0.768 \pm 0.010$ | $0.670 \pm 0.020$ | $0.033 \pm 0.004$ |
| | GEMCONT (strict SNP panel) | $0.763 \pm 0.011$ | $0.667 \pm 0.021$ | $0.029 \pm 0.003$ |

## 3.5. Implementation Details

All models were implemented in PyTorch (Paszke et al., 2019) and trained for 150 epochs with a batch size of 1024 using AdamW (Loshchilov and Hutter, 2017) (base learning rate $3 \times 10^{-4}$, weight decay $1 \times 10^{-4}$) and a cosine-annealing schedule with a 10-epoch warm-up. For all models and experiments, we set the embedding dimension to $D = 256$ and used a single linear layer as the phenotype projector (output dimension $\mathbb{R}^{D/2}$). Embeddings are $\ell_2$-normalized before computing similarities. The temperature $\tau$ in the multimodal contrastive loss is implemented as a learnable scalar (initialized to $\tau_0 = 0.07$). For the supervised baselines we additionally applied early stopping on the validation loss with a patience of 50 epochs. To control for population structure, we regress out the top 20 genetic PCs from both single-variant dosages and PRS features before training, following standard UK Biobank practice (Bycroft et al., 2018; Canela-Xandri et al., 2018). For the genetic branch, weights connected to each input variant were initialized to the corresponding effect size from the external meta-analytic GWAS, and genetic inputs were augmented using SCARF (Bahri et al., 2021) with corruption probability $p = 0.1$. Image augmentations were adapted to each experiment: random erasing for the EMNIST simulation to preserve the rotation signal, random resized crops with Gaussian blur for spirometry, and random resized crops, color jitter, and Gaussian blur for fundus images. Training was performed on a single NVIDIA H100 (80GB) GPU; a typical GEMCONT run (150 epochs, batch size 1024) takes $\sim$10 hours wall-clock, with runtime dominated by the image branch (particularly in fundus).

## 4. Conclusion and Future Work

We introduced GEMCONT, a genetics-based multimodal contrastive learning framework that aligns genotype and imaging embeddings to emphasize disease-relevant variation. By leveraging disease-associated genetic variants as supervision, GEMCONT learns imaging representations predictive of future disease risk, positioning genetics as a biologically grounded

supervisory signal for medical imaging. The framework contributes directly to the medical imaging domain by producing disease-predictive embeddings under genetic supervision.

Our findings reinforce a central goal of imaging genetics: identifying intermediate imaging biomarkers that mediate the relationship between genetic variation and disease (Elliott et al., 2018; Meyer et al., 2020). In this context, GEMCONT operationalizes this principle by coupling disease-associated variants with phenotypic representations, guiding the learned imaging features toward mechanistic axes of disease risk. This extends our earlier work on genetics-supervised biomarker discovery (Sens et al., 2024). While as expected only a modest subset of the disease-associated variants is mediated through the imaging modality under study, each recovered locus represents a testable hypothesis linking genetic variation to an interpretable phenotypic feature.

Empirically, GEMCONT was evaluated across datasets of increasing complexity—from a multimodal MNIST benchmark to spirometry-derived flow–volume curves for asthma and retinal fundus photographs for glaucoma—demonstrating robust predictive performance across modalities. In particular, the glaucoma experiment highlights the practical relevance of GEMCONT within a canonical medical-imaging setting and its ability to refine foundation-model embeddings through disease-specific, genetics-based fine-tuning. Methodologically, we confirmed two core design hypotheses: (i) a linear genetic encoder effectively captures additive genotype–phenotype relationships without measurable gains from additional non-linear modeling; and (ii) performance remains stable under a stricter genome-wide significance threshold, indicating robustness to variant inclusion criteria.

Despite these strengths, several limitations remain. First, although ancestry-related biases were mitigated by regressing out genetic principal components and restricting analyses to unrelated individuals of homogeneous ancestry, residual population or site effects may persist despite covariate adjustment, and disentangling these confounders remains an open challenge for multimodal contrastive frameworks. Second, while the comparison between GEMCONT and the PRS baseline used identical variant panels to isolate the effect of individual variants versus aggregate modeling, future work will extend benchmarking to broader variant sets and polygenic scores from genetically correlated traits. Third, uncertainty in GWAS summary statistics was not explicitly modeled, and incorporating uncertainty-weighted variant selection represents a principled avenue for future development. Finally, although the fixed embedding dimensionality of $D = 256$ yielded stable results across all experiments, a systematic exploration of the trade-off between latent dimensionality and model performance across modalities will be valuable for further optimization.

Looking ahead, integrating generative decoders with GEMCONT represents a promising direction to enhance interpretability and facilitate imaging biomarker discovery. Building on emerging frameworks (Chaudhary et al., 2026, 2025; Shilova et al., 2025), such extensions could enable direct visualization of variant-driven imaging changes by decoding along genetic directions in the latent space. Beyond interpretability, future work will extend GEMCONT to additional imaging modalities (e.g., brain MRI) and integrate it within genetic causal inference frameworks (Davey Smith and Hemani, 2014; Sens et al., 2024). Through these developments, GEMCONT will advance the broader goal of genetics-informed imaging by linking genetic variation to intermediate phenotypes that mediate disease risk, thereby supporting biomarker discovery and patient stratification.

## Acknowledgments

This research has been conducted using the UK Biobank Resource (Application Number 87065). FPC and DS were funded by the Free State of Bavaria's Hightech Agenda through the Institute of AI for Health (AIH). FPC acknowledges support from the Chan Zuckerberg Initiative (CZI) through the AI Residency Program. DS and LS acknowledge the support of the research school Munich School for Data Science (MUDS). AVD acknowledges support from NIH grant R01EB033773. JAS acknowledges funding from the German Federal Ministry of Education and Research and the Bavarian State Ministry for Science and the Arts under the Munich Centre for Machine Learning (MCML), and from the German Academic Exchange Service (DAAD) under the Konrad Zuse School of Excellence for Reliable AI (RelAI).

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
