# OpenReview forum: "Genetics-based Multimodal Contrastive Learning Enhances Phenotypic embeddings and Boosts Genetic Discovery"
_MIDL.io/2026/Conference — MIDL 2026 Poster_

### Official Review · Reviewer_nRr1 · 2026-01-10

**Confidence:** 5
**Preliminary Rating:** 5

**Summary:**

Genetic variation serves as a stable anchor for disease risk, enabling the discovery of causal mechanisms underlying complex phenotypic traits. This work proposes GEMCONT, a multimodal contrastive learning architecture that utilizes disease-associated variants as supervision to align genetic and phenotypic embeddings in a joint latent space. The method is applied to high-dimensional medical data in two real-world tasks: recovering asthma-related traits from flow-volume spirometry curves and glaucoma features from retinal fundus photography. Evaluations indicate that GEMCONT yields more predictive phenotypic embeddings and higher genetic signal recovery compared to unsupervised approaches (e.g., VAE, SimCLR) or univariate polygenic risk models.

**Strengths:**

1.This work addresses a critical challenge in biomedical AI by aligning genetic and imaging data within a shared latent space, effectively treating medical images as intermediate phenotypes that link genetic variation to disease.
2. he method’s robustness is demonstrated through comprehensive testing, encompassing a controlled simulation and two large-scale real-world applications for asthma and glaucoma using the extensive UK Biobank cohort.

**Weaknesses:**

1. Despite the utility of the joint embedding space, the method lacks a mechanism for precise feature attribution, making it difficult to disentangle the specific contributions of individual genetic loci and local imaging features to the disorder.
2. The reliance on a linear genetic projector relies on the assumption that genetic variance is predominantly additive; however, this design inherently limits the model’s expressivity by excluding non-linear interactions and potential epistatic effects from the learned representation.

**Detailed Comments:**

Can the current framework be adapted to explicitly model and visualize how specific genetic variants causally drive changes in the imaging phenotype, rather than just associating them in a shared space?

**Justification Of The Preliminary Rating:**

The authors present a method utilizing contrastive loss to perform joint imaging-genetics analysis. The model architecture combines a ViT-based image extractor with a linear genetic layer, which is initialized using GWAS effect sizes to map both modalities into a unified space. By utilizing contrastive loss in this latent space, the framework encourages the imaging and genetic features of the same subject to be closer together. The approach was validated on a simulated dataset and two large real-world settings: asthma classification from spirometry curves and glaucoma detection from retinal fundus images. The proposed method achieved the highest performance in all tested scenarios, surpassing comparative baselines.

**Questions To Address In The Rebuttal:**

1. Could you elaborate on the rationale behind the design and selection of the latent dimension $D$? Specifically, how was the optimal dimensionality determined to balance representation capacity with the statistical power required for the downstream association testing?
2. Regarding the genetic branch of the framework, did you investigate the use of non-linear neural networks (e.g., Multi-Layer Perceptrons) as an alternative to the linear projector? If so, how did the performance of a non-linear encoder compare to the proposed linear approach in terms of signal recovery and overfitting?

---

> ### Author Response · Authors · 2026-01-24
>
> We thank the reviewer for the positive assessment and for highlighting the novelty and methodological contributions of our work. We address the main points below:
>
> R3.1) Feature attribution and interpretability. \
> We appreciate this insightful suggestion, which aligns closely with ongoing work in our group. In related frameworks such as HistoGWAS (Chaudhary et al., 2024, bioRxiv) and REECAP (Shilova et al., 2025, medRxiv), generative decoders are trained to invert medical imaging encoders, enabling visualization of image changes associated with individual genetic variants by decoding along specific embedding directions. This strategy is directly applicable to the GEMCONT imaging encoder and would allow visualization of phenotypic changes linked to specific variants or variant combinations. We now discuss this integration as a natural future extension in the revised Discussion (lines 365-374).
>
> R3.2) Latent dimensionality. \
> We fixed the embedding dimension to D = 256 across all experiments and now discuss the exploration of the trade-off between representation capacity and statistical power as an avenue for future work in the revised Discussion (lines 361-364).
>
> R3.3) Linear versus nonlinear genetic projector. \
> In response to this point and related comments from Reviewers 1 and 2 (R1.3, R2.4), we added experiments (lines 164-167) replacing the linear genetic projector with a two-layer MLP. This modification yielded no measurable gain in disease-prediction AUC or genetic-signal recovery (Table 1, lines 294-298), empirically confirming that a linear genetic mapping is sufficient at these sample sizes.

---

> ### Comment · Area_Chair_cPmB · 2026-02-02
> **Please provide final rating**
>
> Dear reviewer,
>
> the rebuttal and discussion periods are over, could you please submit your final rating today ?
>
> To do so please use "edit" on your official review.
>
> Thank you again for your precious help in evaluating the paper.

---

### Official Review · Reviewer_amKr · 2026-01-10

**Confidence:** 3
**Preliminary Rating:** 3
**Final Rating:** 4

**Summary:**

The authors introduce a multimodal contrastive learning framework to align genetic variants with imaging phenotypes. They use CLIP-style contrastive learning. The key innovation is by using disease-associated genetic variants (selected from summary statistics) to guide the representation learning. The authors test their method on UK Biobank data and results suggest improvements over self supervised baselines.

**Strengths:**

1. The paper is well written and easy to follow. The sections flow nicely together.

2. The design choices are principled and well explained.

3. The experimental validation is quite comprehensive across multiple simulations where the ground truth is controlled, using two clinical applications.

4. Covariates are adjusted for and ranks normalised. Overall good methodological rigour.

**Weaknesses:**

1. It seems the genetic signal recovery is low across the board. Although GEMCONT is best, it recovers only 3% of known loci for glaucoma

2. Some key implementation details are not obvious. I can't find the embedding dimension, the architecture for the phenotype projector or the final learned temp parameter. It would be hard to reproduce this work.

3. The comparisons feel incomplete. SimCLR and VAE baselines are missing for fundus images and it's not clear why this is. The PRS baseline also uses the same variants as GEMCONT but only as a weighted sum, it seems a fairer baseline would be a nonlinear PRS only model. It's not clear why this is not the case.

4. There doesn't seem to be any account for the uncertainty in GWAS summary statistics, apart from the hard cutoff at 10^-5. I wonder if accounting for uncertainty with weighting would be appropriate here. There doesn't seem to be any discussion of this.

5. Related to the previous point, there doesn't seem to be a discussion or limitations section at all. The conclusion and future work section is very brief. Acknowledgement and awareness of the limitations of the approach and suggestions for potential alleviations for this (not just adding in more features) would be very useful, and expected.

**Detailed Comments:**

- "more subtler" should probably be "more subtle" or just "subtler"
- GEMCONT is never actually defined as an acronym/abbreviation
- CLIP-loss is used in figure 1 but other places use "multimodal contrastive loss". Consider using consistent terms

**Justification Of Final Rating:**

I thank the authors for their comments. The responses strengthen the paper and form more complete claims, especially for the implementation details, encoder assumption and threshold sensitivity analysis. And finally the limitations section is a valuable addition.

The venue fit question still remains. The authors make good points on the the disease specific representation learning. And to reiterate, I'm not arguing that it's against the CFP, but I feel it's a borderline fit for the MIDL core audience.

Overall, I have increased my score to weak accept.

**Justification Of The Preliminary Rating:**

This is a good paper with quite thorough experiments. Although the margin of improvement is quite small, that doesn't contradict the methodological contribution. What's holding the paper back are two things: 1) missing experimental details and discussion (see weaknesses) and 2) the fact that this is mostly a genetics paper with a thin overlap with medical imaging. The reformatting of 1D spirometry data into images is not traditional medical imaging and the fundus experiments use very standard off-the-shelf models without any changes. The GWAS guided approach is very interesting and there are smaller aspects of the paper that are medical imaging, but it might be better suited for a genetics specific venue where reviewers could more properly evaluate the genetic methodology. I am not against it being published at MIDL and it's not against the CFP, but the venue fit is something to consider given the contributions of the paper.

**Questions To Address In The Rebuttal:**

1. What was the embedding dimension for the experiments? How sensitive are the results to the embedding dim size?

2. Does performance vary with variant selection threshold? I.e. adjusting p<10^-5

3. Why are SimCLR/VAE baselines not included for fundus?

4. What is the computational cost of the method? Time/memory.

---

> ### Author Response · Authors · 2026-01-24
>
> We thank the reviewer for the constructive and detailed feedback. The reviewer raised several points, which we address below.
>
> R2.1) Clarification of genetic-signal recovery levels. \
> We thank the reviewer for the opportunity to clarify this point. The observed recovery fractions are expected, as only a subset of disease-associated variants is mediated through the specific imaging modality under study. Importantly, each recovered locus provides a concrete and testable hypothesis about an imaging marker of an intermediate biological process linking genetic variation to observable phenotype. This aligns with the central goal of imaging genetics—to uncover mechanistic bridges between genotype and phenotype rather than to maximize overlap with all disease-associated loci. We clarify this rationale in the revised Discussion (lines 331-339) and provide supporting context from prior imaging-genetics studies.
>
> R2.2) Implementation details and reproducibility. \
> We thank the reviewer for noting this omission. The implementation details now explicitly specifies all key parameters for reproducibility: embedding dimension, phenotype projector architecture, and temperature parameter (lines 306-309).
>
> R2.3) SimCLR/VAE baselines for fundus data. \
> We thank the reviewer for the opportunity to clarify this point. For the fundus application, we rely on RetFound (Zhou et al., Nature, 2023; Zhou et al., Research Square, 2025), a large-scale self-supervised model pretrained on nearly one million fundus images, which represents the current state of the art for unsupervised representations in retinal imaging. Retraining SimCLR or VAE models from scratch on our smaller glaucoma subset would yield substantially weaker results. As described in the original submission (lines 268-275), we use RetFound’s frozen ViT features with mean pooling as the unsupervised reference and train an attention-pooling layer plus a two-layer MLP for GEMCONT. To ensure that the rationale for using RetFound rather than training from scratch is clear early in the paper, we have added a note in the Compared Methods paragraph of Experiments (lines 173–176).
>
> R2.4) Nonlinear PRS baseline. \
> We intentionally employ linear genetic models in both GEMCONT and the PRS baseline, consistent with extensive evidence from human genetics indicating that common-variant effects are predominantly additive, with limited support for genetic interactions (Hill et al., PLoS Genet., 2008; Palmer et al, Science, 2023). That said, we agree that this assumption warrants empirical validation, as also noted by Reviewer 1 (R1.3). Accordingly, we added experiments replacing the linear genetic projector with a two-layer MLP. This modification yielded no improvement in disease-prediction AUC or genetic-signal recovery (Table 1, lines 294–298), empirically confirming that a linear genetic mapping is sufficient at these sample sizes.
>
> R2.5) Accounting for uncertainty in GWAS summary statistics. \
> We currently treat GWAS summary statistics as point estimates for variant selection, without explicitly modeling uncertainty. Incorporating uncertainty into variant weighting or selection would be a principled extension to improve robustness and interpretability. We now acknowledge this limitation and discuss it as a potential avenue for future work in the revised Discussion (lines 358-360).
>
> R2.6) Sensitivity to the embedding dimension. \
> We use a fixed embedding dimension of D = 256 across all experiments (see revised Implementation Details, lines 306–309). While we did not conduct a dedicated sweep over D in this submission, we now explicitly note this as a current limitation and an avenue for future work in the revised Discussion (lines 361-364).
>
> R2.7) Discussion and limitations. \
> We thank the reviewer for this helpful comment. While the original submission was space-constrained, we have now expanded the Conclusion and Future Work section to provide a more comprehensive interpretation of the results and an explicit reflection on methodological limitations. The revised version includes dedicated discussion of confounding factors, architectural design choices, and potential future extensions, addressing the reviewer’s concerns regarding completeness and depth (lines 325-374).
>
> R2.8) Computational cost. \
> We trained GEMCONT on a single NVIDIA H100 (80 GB) GPU with a batch size of 1024. Under this setup, a typical training run of 150 epochs requires approximately 10 hours of wall-clock time. The genetic branch is linear and adds negligible computational overhead; runtime is dominated by the imaging backbone, particularly for the fundus application. We now include these details in the revised Implementation Details (lines 320-322).
>
> (Please see below for R2.9-R2.11)

---

> > ### Author Response · Authors · 2026-01-24
> >
> > (Please see above for R2.1-R2.8)
> >
> > R2.9) Variant-selection threshold \
> > In the initial submission, we selected disease-associated variants from external GWAS using a standard clumping-and-thresholding scheme with a significance threshold of p < 1×10⁻⁵, consistent with prior work (Choi et al., Nat. Protoc., 2020). We agree that it is valuable to assess sensitivity to this choice. To this end, we re-ran GEMCONT using a stricter genome-wide significance threshold (p < 5×10⁻⁸) within the same clumping-and-thresholding procedure. Across both asthma and glaucoma, the stricter panel yielded slightly lower disease-prediction AUCs and no consistent improvement in genetic-signal recovery. These analyses are now introduced in Experiments (lines 165–167), with results presented in Table 1 and discussed in lines 299–301.
> >
> > R2.10) Venue fit. \
> > GEMCONT is a disease-specific representation-learning framework for medical imaging, in which disease genetics provides supervision to learn imaging embeddings predictive of future disease. The scope aligns directly with MIDL’s focus on deep learning for medical imaging and multimodal biomedical data. We evaluate GEMCONT across datasets of increasing complexity (MNIST → spirometry → fundus), with the glaucoma application illustrating the method’s relevance in a fully realized, canonical medical-imaging setting. The positioning of GEMCONT within the medical-imaging domain is now made explicit in the revised Abstract (lines 10–12), Introduction (lines 43–49), and Discussion (lines 325-329).
> >
> > R2.11) Editorial corrections. \
> > We thank the reviewer for the careful reading and helpful suggestions. We corrected the grammatical error (“more subtler” → “more subtle,” line 39), defined GEMCONT as an acronym at its first mention (lines 8–9, 41–42), and standardized terminology throughout, using multimodal contrastive loss consistently in place of “CLIP-loss”. We greatly appreciate the reviewer’s attention to these details.

---

### Official Review · Reviewer_BZBM · 2026-01-17

**Confidence:** 2
**Preliminary Rating:** 3
**Final Rating:** 5

**Summary:**

The authors propose GEMCONT, a multimodal contrastive learning framework designed to align genetic data with medical imaging phenotypes in a shared latent space. The authors propose to use a linear genetic projector to encode genotype data, motivated by the biological prior that genetic effects on complex traits are predominantly additive. This is contrasted with a deep neural network (e.g., ViT or CNN) used for the imaging arm.

The method is validated on a simulated dataset and two large-scale real-world applications using UK Biobank data: spirometry (flow-volume curves) for asthma and retinal fundus images for glaucoma. The authors demonstrate that GEMCONT embeddings outperform standard unsupervised baselines and polygenic risk scores in downstream disease risk prediction and in the recovery of genetic association signals.

**Strengths:**

- Using genetics as a upstream signal to bias representation learning toward risk-relevant phenotypic variation is well-motivated, and the CLIP-style formulation is straightforward and implementable at scale (which is important nowdays).

- The paper goes beyond toy datasets to validate the method on two distinct, large-scale clinical scenarios in the UK Biobank (Asthma/Spirometry and Glaucoma/Fundus).

- The use of a linear genetic projector is consistent with additive genetic architectures and supports interpretability of variant contributions.

**Weaknesses:**

- While the manuscript references prior multimodal approaches, I would appreciate further clarification on how GEMCONT is positioned relative to them. Specifically, are the differences primarily in the architectural choices or the training objectives? Adding a direct comparison or a discussion would significantly improve the clarity of the paper's contribution.

- Given the subject-level alignment, there is a risk that embeddings capture confounders (e.g., population structure, site effects) rather than disease pathology. It would be valuable to discuss how the framework disentangles these factors from true phenotype-genotype connections.

**Detailed Comments:**

Please see weakness

**Justification Of Final Rating:**

The empirical results across simulation and the two large-scale UK Biobank settings remain impressive, and the added clarifications on confounding controls reduce several of my earlier concerns. The initial score is relatively low is because lack of experience in this field. The authors reply makes everything more clear.

**Justification Of The Preliminary Rating:**

**Why not higher?** The core claim that a linear genetic projector is superior to deep encoders lacks empirical proof; an ablation (Linear vs. MLP) is missing. Additionally, the method is primarily benchmarked against unimodal baselines (SimCLR/VAE), making it unclear how it compares to standard multimodal architectures like ContIG.

**Why not lower?** The framework is conceptually distinct and biologically well-motivated. The authors demonstrate clear utility on two large-scale real-world datasets (UK Biobank Asthma & Glaucoma), proving that adding genetic supervision effectively refines phenotypic embeddings compared to standard unsupervised methods.

Please note that my expertise lies in deep learning and representation learning. I am not familiar with the specific protocols for genetic quality control or the biological significance of the SNP selection. I defer to other reviewers on these domain-specific aspects and restrict my assessment exclusively to the proposed model architecture and validation methodology.

**Questions To Address In The Rebuttal:**

Please see weakness 2

---

> ### Author Response · Authors · 2026-01-24
>
> We thank the reviewer for the constructive and thoughtful feedback and for the overall positive assessment of our work. We address the main points raised below:
>
> R1.1) Positioning relative to prior multimodal methods. \
> The key distinction between GEMCONT and prior multimodal frameworks such as ContIG and MRM lies in the training objective and the use of genetics. While ContIG and MRM use genetic or molecular data for task-agnostic multimodal pretraining followed by downstream fine-tuning, GEMCONT directly learns disease-specific imaging embeddings, supervised by disease-associated genetic variants. These embeddings capture imaging biomarkers predictive of future disease risk. This formulation extends our earlier work on genetics-supervised biomarker discovery (Sens et al., Genome Res., 2024). We have clarified this distinction in the Abstract (lines 10–12) and Introduction (lines 43–49).
>
> R1.2) Control for confounding. \
> Population structure constitutes the principal confounder in imaging-genetics analyses. To minimize these spurious correlations between genotype and phenotype, we regress out the top 20 genetic principal components (PCs) from all genotype features prior to training, following standard procedures in UK Biobank analyses (Bycroft et al., Nature, 2018; Canela-Xandri et al., Nat. Genet., 2018). The same PCs, together with age, sex, body-mass index, genotyping array and assessment center, are included as covariates in all downstream embedding–GWAS analyses. Site and scanner effects are unlikely to produce systematic cross-modal correlations—except indirectly through shared population structure—and therefore do not yield consistent alignment signals under the contrastive objective. These clarifications have been incorporated into Methods (lines 126–128), Implementation details (lines 311–313), and Discussion (lines 351-355).
>
> R1.3) Linear vs. nonlinear genetic projector. \
> To evaluate whether additional nonlinear modeling capacity improves performance, we implemented a two-layer MLP (BatchNorm + ReLU) as a drop-in replacement for the linear genetic projector. Across both the asthma and glaucoma experiments, the MLP showed no statistically significant improvement in AUC or genetic-signal recovery. This finding supports that a linear genetic mapping is sufficient for capturing the relevant genotype-phenotype relationships. These analyses are now described in Methods (lines 165–167), with results summarized in Table 1 and discussed in lines 294–298.
>
> R1.4) Comparison with multimodal baselines. \
> GEMCONT was benchmarked against a ContIG-like multimodal baseline in which the genetic branch receives as input a disease-specific polygenic-risk-score (PRS) module using the same set of disease-associated variants. This setup isolates the benefit of modeling individual variant contributions (as in GEMCONT) versus a single aggregate genetic predictor (as in the PRS baseline). We intentionally did not include broader genetic signals (e.g., variants from related traits), which would be feasible for both models, to ensure that both operate under identical variant panels and to attribute performance differences specifically to the use of individual variants versus an aggregate PRS. We now clarify this rationale and note broader benchmarking as a direction for future work in the revised Discussion (lines 355-358).

---

> > ### Comment · Reviewer_BZBM · 2026-01-31
> >
> > Thank you for the thorough rebuttal and for making the paper’s positioning clearer. I now better understand how GEMCONT differs from prior multimodal frameworks.
> >
> > The new linear-vs-MLP ablation is particularly helpful—showing that a simple linear genetic projector is sufficient is not a trivial design choice but a non-trivial empirical statement that strengthens the methodological claim.
> >
> > Overall, the empirical results across simulation and the two large-scale UK Biobank settings remain impressive, and the added clarifications on confounding controls reduce several of my earlier concerns. Based on these updates, I will increase my score.

---

### Author Rebuttal · Authors · 2026-01-24

**Rebuttal:**

We thank all reviewers for their thoughtful and constructive feedback. We appreciate the positive assessment of GEMCONT’s novelty, methodological clarity, and experimental validation, as well as the helpful suggestions for improving the paper.

In response, we submitted a revised manuscript together with a detailed point-by-point rebuttal. Each response references the corresponding line changes, and modifications in the manuscript are annotated to indicate how they relate to the reviewers’ comments.

Key updates include:

1) Robustness analyses: we compared the linear genetic projector with a nonlinear MLP alternative to evaluate model expressivity and confirm the adequacy of the linear design.

2) Technical details and clarifications: expanded description of hyperparameters, architectural components, and computational requirements to improve reproducibility.

3) Discussion and limitations: extended discussion of confounding factors, variant-selection uncertainty, latent dimensionality, and potential directions for interpretability and causal analysis.

We have carefully considered the reviewers’ comments and incorporated them where possible. We believe the revised manuscript clarifies the methodological contributions and strengthens the empirical evaluation. We sincerely thank the reviewers for their insightful feedback, which significantly improved the clarity and rigor of this work.

**Supporting Material:**

/attachment/4acb3a6aada6323e671de0e2e48af74a63316a1e.pdf

---

### Comment · Area_Chair_cPmB · 2026-01-26
**Post-rebuttal discussion and final ratings**

Dear reviewers,

Thank you for providing your comments on the paper. The authors have replied, and possibly modified their paper as a result.

You now have until the 1st of February to discuss with authors via the forum, and provide your final rating. This can change or stay the same depending on the rebuttal and discussion.

Your review, discussion, and final rating will be taken into account for the meta-review.

Once again thank you very much for your help.

---

### Meta-Review · Area_Chair_cPmB · 2026-02-05

**Recommendation:** Accept (Oral)
**Confidence:** 4

**Metareview:**

No substantial issues remain after rebuttal, and reviewers raised their scores.
If we are to make progress towards precision medicine, it is very useful to have methods to deal with data beyond imaging. This represents a promising advance.

---

### Decision · Program_Chairs · 2026-02-13

Accept (Poster)